# Polygenic risk for autism spectrum disorder associates with anger recognition in a neurodevelopment-focused phenome-wide scan of unaffected youths from a population-based cohort

**Frank R. Wendt**[1], **Carolina Muniz Carvalho**[1,2], **Gita A. Pathak**[1], **Joel Gelernter**[1,3], **Renato Polimanti**[1]*

**1** Department of Psychiatry, Yale School of Medicine and VA CT Healthcare Center, West Haven, United States of America, **2** Department of Psychiatry, Universidade Federal de São Paulo (UNIFESP), São Paulo, SP, Brazil, **3** Departments of Genetics and Neuroscience, Yale University School of Medicine, New Haven, United States of America

* renato.polimanti@yale.edu

## Abstract

The polygenic nature and the contribution of common genetic variation to autism spectrum disorder (ASD) allude to a high degree of pleiotropy between ASD and other psychiatric and behavioral traits. In a pleiotropic system, a single genetic variant contributes small effects to several phenotypes or disorders. While analyzed broadly, there is a paucity of research studies investigating the shared genetic information between specific neurodevelopmental domains and ASD. We performed a phenome-wide association study of ASD polygenetic risk score (PRS) against 491 neurodevelopmental subdomains ascertained in 4,309 probands from the Philadelphia Neurodevelopmental Cohort (PNC) who lack an ASD diagnosis. Our main analysis calculated ASD PRS in 4,309 PNC probands using the per-SNP effects reported in a recent genome-wide association study of ASD in a case-control design. In a high-resolution manner, our main analysis regressed ASD PRS against 491 neurodevelopmental phenotypes with age, sex, and ten principal components of ancestry as covariates. Follow-up analyses included in the regression model PRS derived from brain-related traits genetically correlated with ASD. Our main finding demonstrated that 11-17-year old probands with the highest ASD genetic risk were able to identify angry faces ($R^2$ = 1.06%, p = 1.38 × $10^{-7}$, $p_{Bonferroni-corrected}$ = 1.9 × $10^{-3}$). This ability replicated in older probands (>18 years; $R^2$ = 0.55%, p = 0.036) and persisted after covarying with other psychiatric disorders, brain imaging traits, and educational attainment ($R^2$ = 0.2%, p = 0.019). We also detected several suggestive associations between ASD PRS and emotionality and connectedness with others. These data (i) indicate how genetic liability to ASD may influence neurodevelopment in the general population, (ii) reinforce epidemiological findings of heightened ability of ASD cases to predict certain social psychological events based on increased systemizing skills, and (iii) recapitulate theories of imbalance between empathizing and systemizing in ASD etiology.

**Data Availability Statement:** All relevant data are within the manuscript and its Supporting Information files.

**Funding:** This study was supported by the Simons Foundation Autism Research Initiative (SFARI Explorer Award: 534858) and the American Foundation for Suicide Prevention (YIG-1-109-16). CMC was supported by a Fundação de Amparo à Pesquisa do Estado de São Paulo (FAPESP 2018/05995-4) international fellowship. Support for the collection of the data for Philadelphia Neurodevelopment Cohort (PNC) was provided by grant RC2MH089983 awarded to Raquel Gur and RC2MH089924 awarded to Hakon Hakonarson. Support for the collection of PNC data was provided by grant RC2MH089983 awarded to Raquel Gur and RC2MH089924 awarded to Hakon Hakonarson. The funders had no role in study design, data collection and analysis, decision to publish, or preparation of the manuscript.

**Competing interests:** The authors have declared that no competing interests exist.

## Author summary

Large-scale genetic studies have identified many regions of the genome associated with autism spectrum disorder that are considered common in the general population. We investigated how the additive effects of these genetic variations associate with neurodevelopment in youths who lack an ASD diagnosis to better understand how genetic risk for ASD may contribute to other aspects of mental health. We uncovered a relationship between greater genetic risk for ASD and more accurate recognition of angry emotions in others, which persists after considering genetic associations with other psychiatric disorders, educational attainment, and brain region volume. This finding is consistent with existing theories of the relationship between ASD genetic liability and a person's ability to build generalizable and impulse driven models for responding to social phenomena.

## Introduction

Autism spectrum disorder (ASD) describes a group of pervasive neurodevelopmental disorders characterized by impaired social and communication skills. ASD typically manifests as a heterogeneous combination of repetitive and restrictive behaviors along with intellectual capabilities ranging from above average intelligence quotient to intellectual disability [1]. ASD affects between 1–1.5% of the general population and is diagnosed more frequently in males than females [2]. ASD often causes serious social impairment for affected individuals, although some function quite normally.

The polygenic nature of ASD and the relatively common frequencies of most ASD risk alleles pose several population-level questions. For example, why genetic liability to ASD correlates with better cognitive function [3]. We might suspect that the effects of ASD risk alleles in unaffected individuals might relate to social cognition, psychopathology, executive control, memory, and/or sensorimotor capabilities. Here we investigate which of these features are associated with the genetic risk for ASD.

Grove, *et al.* [1] reported an estimated SNP-based observed-scale ASD heritability of ~12% in Europeans. There were robust genetic and phenotype correlations between ASD, cognitive ability, educational attainment, and several behavioral traits [1]. The highly polygenic nature and relatively high contribution of common genetic variation to ASD allude to a high degree of pleiotropy between ASD and other psychiatric and behavioral traits. In a pleiotropic system, a single genetic variant contributes small effects to several phenotypes or disorders. While analyzed broadly, there is a paucity of research studies investigating the shared genetic information between specific neurodevelopmental domains and ASD. Under the hypothesis that polygenic risk for ASD is associated with features of neurotypical development, we attempt to identify these domains in the present study. One way to identify such relationships is a phenome-wide association study (PheWAS). PheWAS are a way to statistically evaluate the pleiotropic nature of a risk variant in relation to other similar or disparately related phenotypes [4–8]. The PheWAS approach to characterizing pleiotropy also has been applied to investigate how the polygenic architecture of a phenotype associates with other phenotypes [5–8]. To our knowledge, phenome-wide scans of ASD polygenic risk has not been performed previously.

Here we conducted a phenome-wide scan, testing the relationship between ASD polygenic risk and hundreds of neurodevelopmental phenotypes in young probands from the Philadelphia Neurodevelopmental Cohort (PNC) without an ASD diagnosis [9–12]. Based on high-resolution polygenic risk scoring (PRS), we observed a significant positive association between

ASD PRS and the ability to recognize anger. Conversely, we report a suggestive negative relationship between ASD PRS and the ability to correctly distinguish the age of others. These results together with other suggestive associations highlight several features of human neurodevelopment in the young, such as emotional intelligence, as key pathophysiological targets for ASD etiology.

## Results

An overview of the methodological approach used in this study is provided in Fig 1.

### PNC participant inclusion and description

The Philadelphia Neurodevelopmental Cohort (PNC) consists of youths aged 8–21 stratified by PNC study-organizer-defined age groups (Table 1). PNC was designed to study the genomics of complex pediatric disorders but is not enriched for any specific disorder; the cohort is considered generally healthy. All participants underwent clinical assessment, including a neuropsychiatric structured interview and review of electronic medical records. They were also administered a neuroscience-based computerized neurocognitive battery and a subsample underwent neuroimaging.

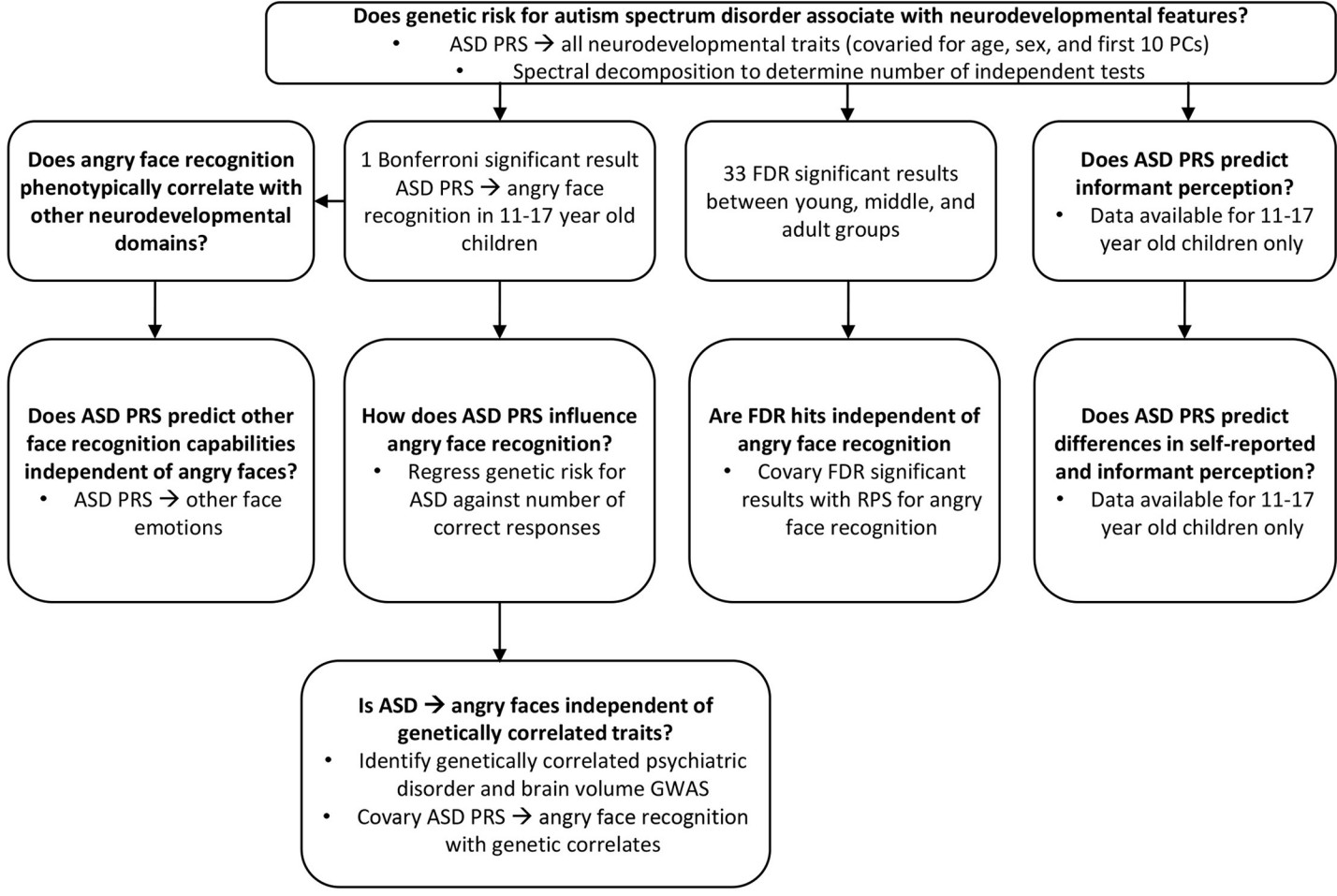

**Fig 1. Analysis flowchart with relevant research questions in bold text.**

**Table 1. PNC sample description.** Samples sizes for each age group (defined by Philadelphia Neurodevelopmental Cohort) after quality control and selection of unrelated European samples.

| Proband | Age, years | Male N | Female N | Total N | Phenotype N |
|---|---|---|---|---|---|
| Young | 8–10 | 579 | 456 | 1,035 | 311 |
| Middle | 11–17 | 1,197 | 1,302 | 2,499 | 491 |
| Adult | 18 and over | 345 | 430 | 775 | 324 |

This study aimed to discover attributes of neurodevelopment associated with genome-wide additive risk for ASD. To avoid associations confounded by the presence of ASD cases in the target data, PNC probands were excluded from this study if they or their informant answered "yes" to the question "Autism or Pervasive Developmental Disorder—Do/did you have this problem?" (MED291), resulting in removal of fewer than 5% of each proband group [13]. Note that the PNC ASD probands are insufficiently powered to detect an association between ASD PRS and PEITANG (N = 184).

All study participants between the ages of 8 and 18 provided assent and parental approval to participate in the PNC. Probands over 18 years of age provided consent to PNC participation.

## Neurodevelopmental trait prediction

ASD PRS was regressed against 491 neurodevelopmental phenotypes in youths lacking an ASD diagnosis. These participants were analyzed in three age groups defined in the data structure of the PNC (young, middle, and adult; Table 1) and consistent with age-of-diagnosis statistics for ASD in the United States (young proband, N = 1,035, age 8–10; middle probands, N = 2,499, age 11–17; and adult probands, N = 775 age $\geq$18) (Fig 2, Table 1, and S1 Table). Note that age groups defined within the PNC data structure reasonably correspond to different neurodevelopmental timeframes such as early childhood or middle-childhood to adolescence. To maintain consistent nomenclature with the PNC reporting, we refer to age-stratified cohorts by the PNC-assigned proband grouping.

ASD PRS was significantly associated with the emotional intelligence phenotype PEITANG (Penn Emotional Identification Test (PEIT) recognition of angry emotions; z-score = 5.28, $R^2$ = 1.06%, $p$ = 1.38 × $10^{-7}$, P-value threshold = 0.4487, $N_{SNPs}$ = 51,452; Fig 2) after applying Bonferroni multiple-testing correction accounting for the number of phenotypes and PRS tested (p = 0.05/13,787 independent tests = 3.63 × $10^{-6}$) in subjects aged 11–17 years. The PEIT measures a proband's ability to recognize and identify a face as either happy, sad, angry, fearful, or neutral in a multiple-choice testing format normalizing the age, sex, and race of faces shown to each proband [14]. The PNC reports PEIT accuracy and speed for each emotion individually and all emotions combined as separate phenotype items. The association of ASD PRS with PEITANG in the middle proband group nominally replicated with adult probands of mean age 19 ± 1.2 years (z-score = 2.10, $R^2$ = 0.55%, p = 0.036). Combining the three age-stratified samples, a stronger association of ASD PRS with PEITANG phenotype was observed (z-score = 5.69, $R^2$ = 0.70%, p = 1.37 × $10^{-8}$). When binned by quartiles, there was a positive relationship between increased ASD PRS and the ability to recognize angry faces. The highest quartile of ASD PRS distribution had a 53% increase in correct recognition of angry emotions relative to lowest quartile ($\beta_{q4vs.q1}$ = 0.427, $p_{q4vs.q1}$ = 1.42 × $10^{-6}$; Fig 2).

The *PEITANG* trait was at least nominally significantly correlated with all other emotion recognition accuracies (0.007 $\leq r^2 \leq$ 0.378, 2.20 × $10^{-16} \leq$ p $\leq$ 1.86 × $10^{-5}$) and reaction times in happy, sad, angry, and fearful emotion trials (0.001 $\leq r^2 \leq$ 0.007, 2.87 × $10^{-8} \leq$ p $\leq$ 0.038; S2 Table). To evaluate the relationship between ASD PRS and other emotion recognition tasks independent of anger recognition accuracy, PEITANG was included as a covariate in these

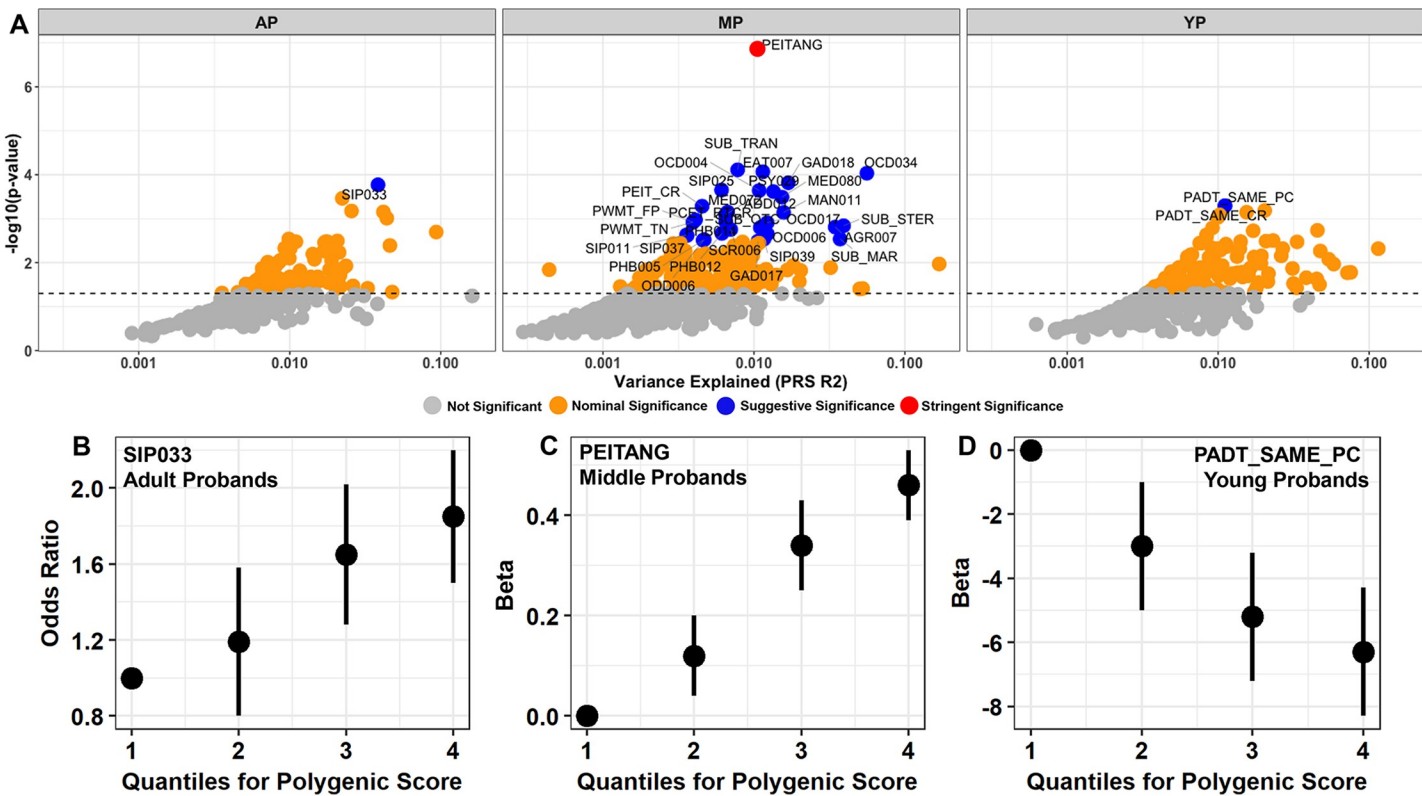

**Fig 2. Autism PRS association with PNC phenotypes.** Overview of best-fit models for autism spectrum disorder (ASD) associating with neurodevelopmental phenotypes in adult (AP, $N_{phenotypes} = 324$), middle (MP, $N_{phenotypes} = 490$), and young (YP, $N_{phenotypes} = 311$) probands of the Philadelphia Neurodevelopmental Cohort. (A) Maximum phenotypic variance explained ($R^2$) by the best model fit for each trait given genetic liability to ASD. The dashed horizontal line represents the nominal significance threshold. (B-D) The relationship between binned ASD polygenic risk scores for AP, MP, and YP participants and the most significant phenotype predicted by ASD genetic liability: (B) SIP033 Structural Interview for Prodromal Symptoms: "Has anyone pointed out to you that you are less emotional or connected to people than you used to be?"; (C) PEITANG: Number of correct responses to anger trials during completion of The Penn Emotional Identification Test (PEIT) for recognizing angry emotions; (D) PADT_SAME_PC: Percent of correct responses to test trials with no age difference during completion of the Penn Age Differentiation Test for detecting which face in a face pair appears older. Note the lowest quartile in figures B-D represents the referent.

models. In models covaried for PEITANG, ASD PRS was significantly associated with the ability to recognize happiness (z-score = -2.41, $R^2$ = 0.221%, p = 0.016, $p_{diff} = 6.41 \times 10^{-4}$) and neutral emotions (z-score = -2.05, $R^2$ = 0.159%, p = 0.041, $p_{diff}$ = 0.433; Fig 3). ASD PRS was associated with reaction time for anger trials (z-score = -2.08, $R^2$ = 0.138%, p = 0.038), fear trials (z-score = -2.06, $R^2$ = 0.161%, p = 0.018), and happy trials (z-score = 2.21, $R^2$ = 0.356%, p = 0.027; Fig 3); however, covarying with PEITANG did not significantly change the relationship between ASD PRS and reaction times for any trials (p>0.05). A related PNC trait, the Penn Emotion Differentiation Test (PEDT), detects a participant's ability to decide which of two faces is more severe. All PEDT trials were weakly associated with ASD PRS (S1 Fig).

Traits associated with ASD PRS using a suggestive threshold are detailed in the S1 Text, S2 and S3 Figs, and S3 Table.

## Influence of related brain traits

We next tested whether the association between ASD PRS and PEITANG was attributed to the shared genetic liability of ASD with other brain-related phenotypes. To do this we considered the association between PEITANG and PRS calculated from brain imaging phenotypes [15], other psychiatric disorders (*i.e.*, attention deficit hyperactivity disorder (ADHD) [16], anorexia

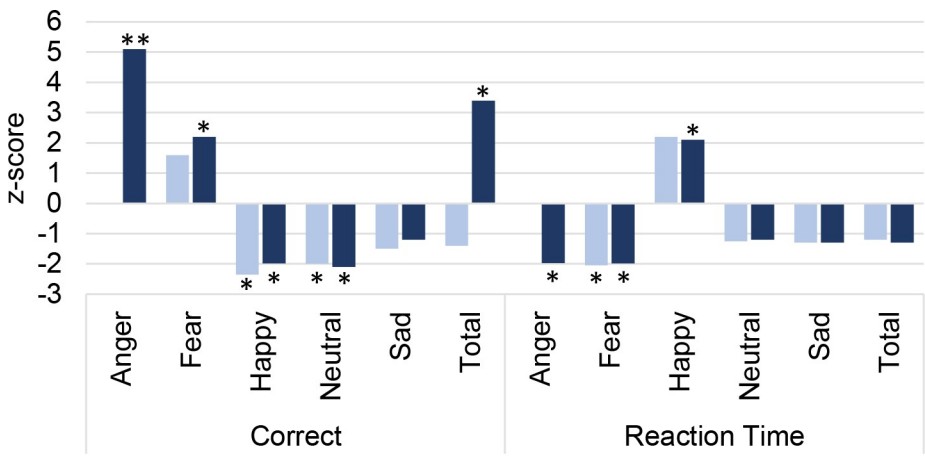

**Fig 3. Predicting emotion recognition with ASD PRS.** Association between facial emotion capabilities of the middle proband group (ages 11 to 17) of the Philadelphia Neurodevelopmental Cohort using polygenic risk (PRS) for autism spectrum disorder. The x-axis shows correctness and response time for each facial emotion using two neurocognitive instruments: The Penn Emotion Identification Test (shaded colors: original results; tinted colors: results covaried for the effects of PEITANG). Significance is indicated by * for $p < 0.05$ and ** for FDR $Q < 0.05$.

nervosa [17], bipolar disorder [18], major depressive disorder [19], Tourette syndrome [20], obsessive compulsive disorder [21], and schizophrenia [22]), and educational attainment [23] (Table 2 and S4 Table). Among these traits, we selected fourteen brain imaging phenotypes and four psychiatric disorders that were significantly genetically correlated with ASD after multiple testing correction [24]. To remove genetically redundant traits, covariates for PRS were selected from genetically correlated trait pairs based on which had the highest SNP-based heritability z-score (S4 Fig). This approach identified six brain imaging phenotypes (left cerebellar white matter volume, mean diffusion tensor mode (MO) in sagittal stratum, mean L1 (*i.e.*, strength of diffusion along the L1 principal axis of diffusion magnetic resonance imaging (dMRI)) in anterior limb of internal capsule (right), mean L3 (*i.e.*, strength of diffusion along the L1 principal axis of dMRI) in uncinate fasciculus (left), right hemisphere central sulcus area, and left hemisphere cuneus gyrus thickness, two psychiatric disorders (ADHD and

**Table 2. Relationship between PEITANG and PRS of additional phenotypes.** Best model fit PRS of psychiatric disorder, educational attainment, and brain image-derived phenotype polygenic risk scores associated with the *PEITANG* phenotype in the Philadelphia Neurodevelopmental Cohort middle proband group (aged 11–17 years old).

| Trait | Unadjusted for ASD PRS | | | Adjusted for ASD PRS | | |
|---|---|---|---|---|---|---|
| | $R^2$ (%) | z-score | p | $R^2$ (%) | z-score | p |
| Schizophrenia | 1.50 | 6.32 | $3.12 \times 10^{-10}$ | 0.898 | 4.89 | $1.05 \times 10^{-6}$ |
| Attention deficit hyperactivity disorder | 0.204 | 2.32 | 0.021 | 0.04 | 1.03 | 0.303 |
| Educational attainment | 0.463 | 3.49 | $4.98 \times 10^{-4}$ | 0.233 | 2.48 | 0.013 |
| Left cerebellum white matter volume | 0.58 | 3.91 | $9.62 \times 10^{-5}$ | 0.318 | 2.9 | $3.35 \times 10^{-4}$ |
| Mean diffusion tensor mode (MO) in sagittal stratum (left) on FA skeleton (from dMRI data) | 0.469 | -3.51 | $4.59 \times 10^{-4}$ | 0.333 | -2.97 | 0.003 |
| Mean L1 in anterior limb of internal capsule (right) on FA skeleton (from dMRI data) | 0.236 | -2.48 | 0.013 | 0.167 | -2.1 | 0.037 |
| Mean L3 in uncinate fasciculus (left) on FA skeleton (from dMRI data) | 0.286 | 2.74 | 0.006 | 0.269 | 2.67 | 0.008 |
| Right hemisphere central sulcus area (from Destrieux Atlas) | 0.56 | -3.84 | $1.27 \times 10^{-4}$ | 0.428 | -3.37 | $7.64 \times 10^{-4}$ |
| Left hemisphere cuneus gyrus thickness (from Destrieux Atlas) | 0.413 | 3.29 | 0.001 | 0.322 | 2.92 | 0.004 |

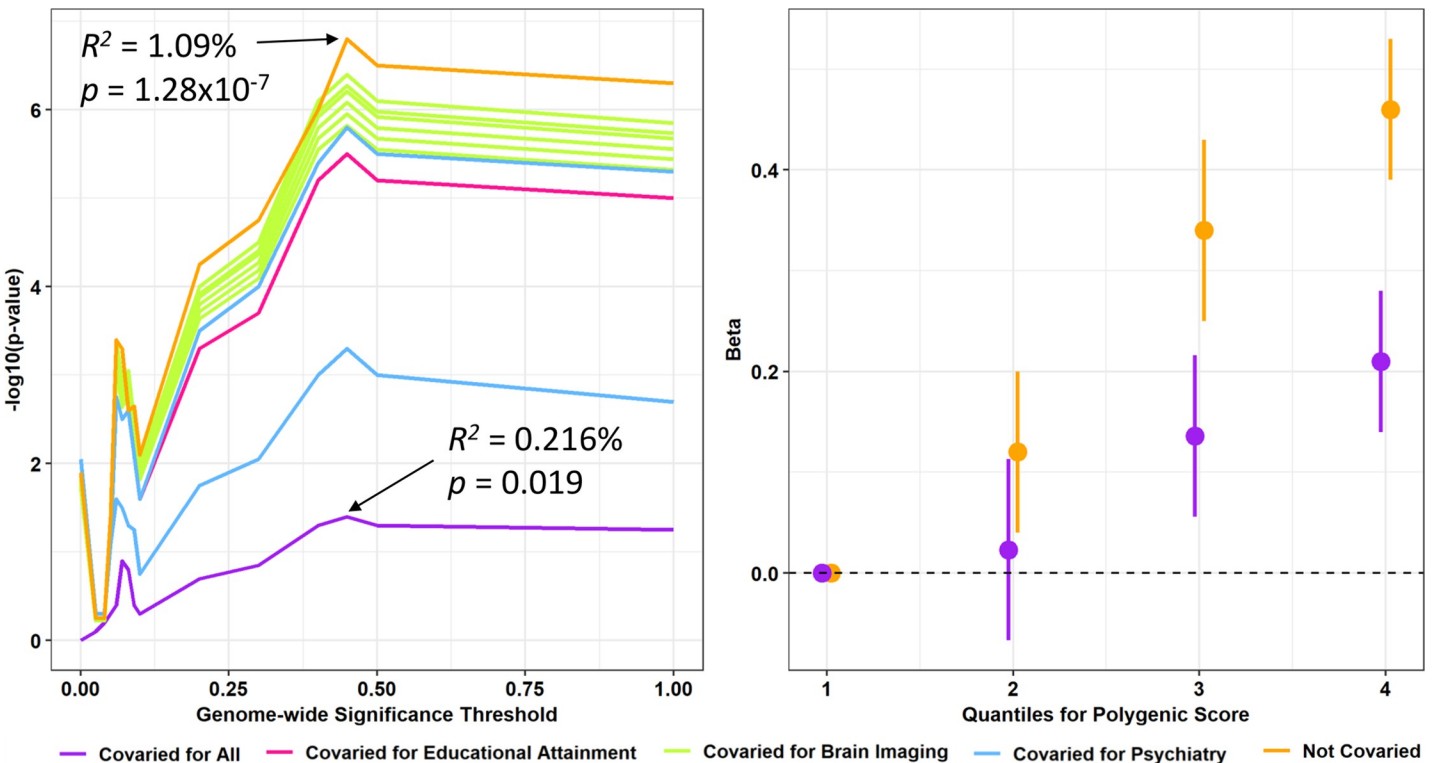

**Fig 4. Robustness of the ASD and PEITANG relationship.** Association between the ability to recognize angry faces in the middle proband group (ages 11–17) using the polygenic risk for autism spectrum disorder (ASD) covaried for age, sex, ten principal components, and the PRS for four brain imaging phenotypes, attention deficit hyperactivity disorder, schizophrenia, and educational attainment. (A) PRS p-value across a range of genome-wide significance (GWS) thresholds; the maximum PRS before and after covarying for all brain and psychiatry traits are labeled. (B) The positive correlation between quartiles of ASD polygenic risk and the number of correct responses to anger recognition trials in the Penn Emotional Intelligence Test; the lowest PRS quartile represents the referent.

schizophrenia), and educational attainment. For complete description of brain image acquisition and brain mapping, refer to Miller, *et al.* 2016 [25] and Elliot, *et al.* 2018 [15]. ASD PRS was significantly associated with the ability to recognize angry emotions in the middle proband group when considering one covariate at a time and when including all covariates in the model (Fig 4). The effects of ASD PRS on the relationship between each covariate and PEITANG also was considered. At matched GWS thresholds, the PRS for schizophrenia, educational attainment, and all six brain imaging phenotypes maintained significant associations with PEITANG after covarying for ASD PRS (Table 2). The relationship between the ADHD PRS and PEITANG was not independent of the effects of ASD PRS.

## Informant perception

Self-reported behavioral attributes (*i.e.*, the person's perception of their own trait or behavior) may deviate from others' perception of that same attribute [26]. In the middle proband age group, we evaluated associations between ASD PRS, informant perception of traits, and the differences between informant-reported and self-reported traits. After multiple testing correction based on the number of phenotypes tested (N = 507; FDR Q < 0.05), 24 informant-reported phenotypes were significantly associated with ASD PRS. The most significantly associated informant-reported phenotype was PHB014 (informant-reported PHB014: z-score = -3.86, $R^2$ = 1.20%, p = $1.18 \times 10^{-4}$) and this relationship agreed with the proband-reported effect direction (PEITANG-covaried proband-reported PHB014: z-score = -3.01, $R^2$ = 0.156%,

p = 0.001, $p_{diff}$ = 3.25 × 10$^{-4}$; S5 Table). The PHB014 trait contains answers to the question "Thinking about all of the time that you were afraid of (insert worst fear), whether or not you actually faced it, how long did this fear last (Months)?"

## Discussion

ASD PRS was at least nominally associated with several aspects of emotion recognition in healthy subjects aged 11 to 17 years old (S5 Fig). Thus, our results tie together the genetic risk for ASD pathology and related traits ascertained in the normal population. We showed that ASD PRS was positively associated with the ability to recognize angry faces (*i.e.*, PEITANG). Emotion recognition deficits are typically considered hallmark attributes of ASD cases [27] although this is often contested in the literature on the basis of different diagnostic instrument accuracy, ASD case severity heterogeneity, and eye movement tracking frequency and accuracy [28]. After covarying all other emotion recognition measure with PEITANG, ASD PRS remained associated with poor recognition of happy and neutral faces in others.

We hypothesize that this paradoxical relationship stems from positive associations between ASD and the ability to predict social phenomena more generally. Recently characterized by Gollwitzer and Bargh [29], people perceptive skills (*e.g.*, "People prefer to interact with people who are different than them, rather than similar to them?" True-False) and person perceptive skills (*e.g.*, "This person appears happy when interacting with others." True-False) share some epidemiological correlates but are considered distinct social psychological skillsets. Gollwitzer, *et al*. demonstrated that the inability of ASD cases to perceive individual intention, emotion, and thoughts is uniquely distinct (and indeed oppositely correlated) from the ability of ASD cases to predict social phenomena more generally [30]. Importantly, this observation was observed across data from 104 countries suggesting that culturally-driven social norms do not confound this skill. The authors propose the skill of systemizing as a likely origin for their observations. Systemizing describes the ability to predict generalizable rules and regularities of a system and is positively associated with cognitive ability and cognitive curiosity. Our detected increased ability of individuals with high genetic risk for ASD, but no ASD diagnosis, to recognize negative emotions in others is reflective of their superior ability to build generalizable "if a, then b" rules (*e.g.*, "if the face is red, the brow is furrowed, and the jaw is clenched, then this person is angry"). After covarying for schizophrenia, ADHD, educational attainment, and six brain imaging phenotypes, ASD PRS remained positively associated with the PEITANG trait. This observation suggests that although PRS for other traits (*e.g.*, schizophrenia) independently predicted recognition of angry faces with greater magnitude than ASD PRS, the relationship between ASD PRS and the ability to recognize anger in others is independent of the effects of highly genetically correlated traits with greater sample sizes. PRS for schizophrenia, educational attainment, cerebellar regions, the uncinate fasciculus, and cuneus were positively associated with recognizing angry faces. This is unsurprising given the relationship between education and psychiatric conditions and perhaps reinforces human and animal models of the role of oxytocin, stress, and anger on the brain [31]. This persistent association between genetic risk for several brain phenotypes, including strong association with neuropsychiatric conditions, suggests that anger identification may be a common feature across neuropsychiatry more generally than just ASD. However, most of the effects observed appear to be independent. This suggests a complex scenario where the association of genetic risk across psychiatric disorders with anger recognition is due to independent mechanisms rather than to a common shared pathway. Additionally, there are notable power differences among GWAS used to generate the PRS. Indeed, the stronger effect observed for SCZ PRS is likely due to it being the most informative GWAS dataset used to generate the PRS.

Three previous studies have investigated the genetic relationships between emotional intelligence, cognitive empathy, and ASD [32–34]. These previous analyses relied on ASD GWAS data from substantially smaller sample sizes than those used herein (N = 46,351 *vs.* N = 10,610 in [32–34]). Nevertheless, our data replicate previous nominally significant findings between ASD PRS and negative emotion recognition tasks (*e.g.*, sadness and fear). To our knowledge, this is the first observation of a relationship between ASD PRS (based on a total sample size over 4 times larger than previous investigations) and angry emotion recognition using genetic data in 11-17-year-old youths.

It is intriguing that the strong significant relationship between ASD PRS and PEITANG in 11-17-year-old probands replicated in adult probands but not young probands. According to our power calculation (see Materials and methods), both adult proband and young proband groups have sample sizes large enough to replicate the PEITANG association at nominal significance level ($p < 0.05$), but not considering a multiple testing correction significance level ($p < 3.63 \times 10^{-6}$; required N = 1,070). One possible explanation for lack of replication across development may be related to how sex balance across proband groups distributes the relationship between puberty, hormones, and social cognition. Females are better at emotion recognition tasks than males [35,36] and there is evidence that late-pubescent and post-pubescent individuals are generally better able to recognize emotions than those in early puberty or pre-pubescents (*e.g.*, the PNC young proband group) [37]. Though adjusted for both age and sex, our analysis includes a larger number of females in both the middle (52% female) and adult (55% female) proband groups relative to the young proband group (44% female). The greater proportion of females in adult and middle proband age groups and the earlier onset of puberty in females likely increases the number of late-pubescent and even post-pubescent individuals in the middle and adult proband groups. We hypothesize that in combination, later onset puberty in males relative to females and the greater number of males in the YP group contributed to masking the relationship between ASD PRS and PEITANG. An additional explanation may relate to documented developmental age-specific correlative and polygenic predictive effects of ASD and other psychiatric disorders on social communication deficits [38,39]. PEITANG is not a social communication deficit and the association between ASD PRS and communication deficits appears to have a negative linear relationship with age. Therefore, our findings reinforce that age is an important moderator of the effect of ASD PRS on behavior in unaffected individuals.

This study has four main limitations. First, the PNC evaluates broad aspects of human neurodevelopment and with respect to social cognition only investigates facial stimuli at the exclusion of gesture, vocalization, and complex social situations. Future work may investigate the relationship between ASD PRS and higher resolution measures of social cognition, including those related to people perception. Second, ASD cases were removed prior to PRS but the remaining proband groups likely contain disproportionate ratios of disorders genetically correlated with ASD which may bias the observed predictions in a direction that favors ASD PRS strongly predicting a PNC trait (*i.e.*, false positives). Given the strength of the relationship between ASD PRS and PEITANG, this potential bias likely lies within the phenotypes predicted by ASD PRS with nominal significance (Fig 1). Even though these biases may influence the specific traits predicted by ASD PRS in each proband group, the strongest relationship (ASD PRS and PEITANG) was replicable across age groups. Third, we were not able to find an adequately powered sample independent from the PNC cohort to externally replicated the PEITANG association (see Materials and methods). Though replicated in the PNC adult proband group, our results have not been replicated externally and should be interpreted as such until verified in a sufficiently large cohort of individuals tested with the PEIT. Future work should aim to confirm this relationship in the middle proband age group as well as across

developmental age groups. Fourth, unfortunately we did not identify a publicly available dataset using the PEIT that also included adult participants to investigate how genetic liability to ASD is associated with anger recognition and other neurodevelopmental traits across the lifespan.

In this study, we demonstrated that ASD PRS is associated with the ability to recognize angry faces in ASD-unaffected individuals; however, ASD PRS was not associated with the ability to differentiate the intensity of emotions. These results were observed in a cohort without enrichment for specific neurodevelopmental phenotypes and self- and informant-reported ASD cases were removed from our analyses [13]. These data support further investigation of how ASD risk alleles relate to psychopathology and neurodevelopment in healthy and ASD patients. Considering epidemiological data regarding emotion recognition, future work may investigate how ASD PRS predicts correct recognition and reaction time for emotion recognition and differentiation tasks in ASD cases and the relationship between ASD genetic risk and features of generalized social cognition skills (*e.g.*, social loafing, misattribution, deindividuation, self-serving bias). Additionally, our findings appear to indicate that ASD PRS association is independent from the genetic risk of other psychiatric and brain-related traits. Further studies based on multivariate statistics will be needed to expand this result, characterizing the molecular mechanisms shared across the psychiatric spectrum and those that are disease-specific.

## Materials and methods

### Genetic data

Genome-wide association study (GWAS) summary statistics for ASD (18,382 cases and 27,969 controls of European descent from Grove, *et al.* 2019) and additional psychiatric disorders (S4 Table) were obtained via the Psychiatric Genomics Consortium (PGC; available at https://www.med.unc.edu/pgc/results-and-downloads). GWAS summary statistics for 3,144 brain image-derived phenotypes were obtained from the Oxford Brain Imaging Genetics project (BIG; available at http://big.stats.ox.ac.uk/) [15,25].

Neurodevelopmental phenotype and genotype data for 9,267 youths aged 8–21 were obtained from the PNC (Neurodevelopmental Genomics: Trajectories of Complex Phenotypes, dbGaP phs000607.v3.p2). Comprehensive details of the clinical and cognitive assessments have been reported previously [9–12]. Briefly, the Children's Hospital of Philadelphia (CHOP) and University of Pennsylvania deeply phenotyped youth participants visiting CHOP or an affiliated clinic for routine visit and volunteered to participate in genomic studies of complex pediatric disorders. The cohort is considered generally healthy and is not enriched for any specific disorder, behavior, or trait.

Clinical testing for each participant included (1) GOASSES (a modified version of the Kiddie-Schedule for Affective Disorders and Schizophrenia) to identify timeline of life events, demographic and medical history, Global Assessment of Functioning, and general interviewer observations, (2) a psychopathology symptom and criterion-related assessment of mood disorders, anxiety disorders, behavioral disorders, psychosis spectrum, eating disorders, suicidal thinking and behavior, and treatment history, and (3) an abbreviated form of the Family Interview for Genetics Studies to assess major domains of psychopathology in the proband's first-degree relatives.

A neurocognitive battery was performed for each PNC proband. The battery yields measures of accuracy and speed for domains of executive control functions (abstraction, attention, and working memory), episodic memory (verbal, facial, and spatial processing), social cognition (emotion identification, emotion intensity differentiation, and age differentiation), and

sensorimotor and motor speed. These neurodevelopmental domains are evaluated using the Penn Conditioning Exclusion Test, Penn Continuous Performance Test, Letter N-back Test, Penn Word Memory Test, Penn Face Memory Test, Visual Object Learning Test, Penn Verbal Reasoning, Penn Line Orientation Test, Penn Emotion Identification Test, and Penn Emotion Differentiation Test.

For a complete list of neurodevelopmental domains assessed and the specific features of each domain tested in the neurocognitive battery, see https://www.ncbi.nlm.nih.gov/projects/gap/cgi-bin/study.cgi?study_id=phs000607.v3.p2.

The PNC assigned each participant to a proband age group based on age at recruitment and sampling: young, middle, or adult (Table 1). ASD in the United States often is diagnosed after age 9 and rarely in youth or young adulthood, even if the individual is symptomatic. The PNC-defined age groups coincide with this trend so we maintained the age-stratified architecture of PNC probands and investigated each age group independently (Table 1). Phenotypes were included in our analysis if they had ≥ 500 participants and effective sample size ≥ 50 (*i.e.*, the number of cases for each phenotype necessary to detect an effect weighted by the proportion of the binary categories with respect to that variable).

## GWAS quality control and trait inclusion

Pre-imputation quality control was performed in plink-v1.9 following PGC analysis pipelines specifically designed to handle large-scale datasets consisting of multiple genotyping platforms (see https://sites.google.com/a/broadinstitute.org/ricopili/preimputation-qc). Individuals of European descent were confirmed with genetic information via principle component analysis using SNPs shared across all bead-chips (38,739 SNPs) and the 1000 Genomes Project reference panel for populations with European ancestry (N = 503). For sample pairs with cryptic relatedness (PI-HAT > 0.2), the sample with more informative phenotypes was retained. Imputation was performed for 4,309 unrelated individuals of European ancestry using SHA-PEIT for pre-phasing [40], IMPUTE2 for imputation [41], and the human 1000 Genomes Project Phase 3 as a reference panel. Samples were stratified by proband group assignment in the PNC for PRS analyses (Table 1).

## Polygenic risk scoring and phenome-wide association

PRS based on ASD GWAS summary statistics were regressed against phenotypes in three PNC proband groups. PRS was performed in PRSice v2 [42] with the default settings (*i.e.*, SNPs were clumped in 250-kb windows on either side of target SNPs using clump-$r^2$ and clump-p thresholds of 0.1 and 1, respectively). High resolution PRS were computed using GWAS variants with GWAS association p-values ranging from p = 0.0001 to p = 1 in 0.000005 increments. PRS were weighted by their effect size in the independent GWAS (*e.g.*, ASD). All PRS were covaried for age, sex, and the first ten PCs. The most significant PRS also was covaried using the PRS of UKB BIG brain imaging phenotypes and other psychiatric disorders using the same PRS procedure described above (Table 2 and S4 Table). To select additional phenotypes to include as covariates in our PRS calculation, SNP-based observed-scale heritability and genetic correlation for PGC psychiatric disorders and BIG brain imaging phenotypes were determined using the Linkage Disequilibrium Score Regression (LDSC) method (24). Covariates from significantly genetically correlated trait pairs were selected for PRS based on which phenotype from each trait pair had the higher SNP-based heritability z-score.

To our knowledge there is no sample overlap between PNC participants and those samples included in the GWAS of ASD or other phenotypes used herein.

To evaluate how traits vary with increased ASD polygenic risk, the individual genetic liabilities to ASD were binned into four quartiles and regressed against PNC traits using a generalized linear model including the same covariates as the respective PRS prediction. The phenotype values for each quartile were then compared to the reference quartile one-by-one with quartile status of ASD risk as a predictor of target phenotype in a regression. We applied a multiple testing correction accounting for the number of independent tests performed in each age group, as determined by spectral decomposition of non-parametric correlations (Spearman's rho) between PRS tested and PNC phenotypes [43]. The nominal significance threshold of 0.05 was adjusted for the sum of independent tests from each age group (13,787 independent tests). Additionally, we considered a suggestive significance threshold based on False Discovery Rate (FDR Q = 5%) accounting for the number of phenotypes tested (N = 491).

## Statistical power

The avengeme R package (Additive Variance Explained and Number of Genetics Effects Method of Estimation) was used to estimate the sample size needed to reach an adequate statistical power to observe significant associations surviving a nominal significance threshold ($p < 0.05$) and the multiple testing correction significance threshold ($p < 3.63 \times 10^{-6}$). The power calculation was conducted considering the PEITANG (Penn Emotion Identification Test correct responses to anger trials) phenotype as the reference, because it showed an association with autism spectrum disorder (ASD) polygenic risk score (PRS) that survived multiple testing correction. Based on a weighted effects model (*i.e.*, the number of risk alleles carried are weighted by per-SNP effect estimates), and number of SNPs contributing to the model, a sample of 281 participants is required to achieve 80% power to detect a nominally significant association ($p < 0.05$) between ASD PRS and PEITANG. With the same model specifics, a sample size of 1,070 is required to achieve 80% power to detect a significant association between ASD PRS and PEITANG that survives multiple testing correction ($p < 3.63 \times 10^{-6}$).

## Phenotype correlation

Spearman's correlation between phenotype measurements of the PNC were calculated in R studio using the rcorr function of the Hmisc library. Correlation p-values were adjusted for the number of tests performed using FDR Q < 0.05.

## Supporting information

**S1 Text. Supplemental results–Neurodevelopmental trait association.**
(DOCX)

**S1 Fig. Prediction of facial emotion differentiation capabilities of the middle proband group (ages 11 to 17) of the Philadelphia Neurodevelopmental Cohort using polygenic risk (PRS) for autism spectrum disorder.** The x-axis shows correctness and response time for each facial emotion using two neurocognitive instruments: The Penn Emotion Differentiation Test Correctness and Reaction Time. Significance is indicated by * for p < 0.05.
(DOCX)

**S2 Fig. Fifty-nine traits phenotypically correlated with PEITANG and at least nominally predicted by polygenic risk (PRS) for autism spectrum disorder (ASD) in the Philadelphia Neurodevelopmental Cohort middle proband group (age 11–17).** Neurodevelopmental traits are labeled if the correlation with PEITANG and PRS survive multiple testing correction; these are PEIT_CR: Penn Emotion Identification Test total correct responses for all trials; GAD018: Generalized Anxiety Disorder: Did you feel any of the following physical symptoms

when you worried the most: irritability (feeling easily annoyed)?; PHB012: Specific Phobia: Thinking about all of the time that you were afraid of (insert worst fear), whether or not you actually faced it, how long did this fear last? (Days); PHB014: Specific Phobia: Thinking about all of the time that you were afraid of (insert worst fear), whether or not you actually faced it, how long did this fear last? (Months).
(DOCX)

**S3 Fig. Z-score converted difference in polygenic risk score (PRS) coefficient for 30 phenotypes suggestively predicted by polygenic risk for autism spectrum disorder in the middle proband group of the Philadelphia Neurodevelopmental Cohort.** Two differences were evaluated: (1; circles) the coefficient difference between original best-fit PRS and the same PRS covaried for the effects of PEITANG (*i.e.*, the same genome-wide significance threshold in both instances) and (2; triangles) the coefficient difference between original best-fit PRS and the best-fit PRS after covarying for the effects of PEITANG (*i.e.*, the GWS may be different in each instance). Note that in several instances the covaried best-fit PRS is the same as the matched PRS from the original test. Filled-in shapes indicate statistically significant differences between original and covaried PRS coefficients (false discovery rate Q < 0.05).
(DOCX)

**S4 Fig. Selection of brain image-derived phenotypes as covariates in the polygenic risk score analyses between autism spectrum disorder (ASD) and neuropsychiatric traits in the young.** (A) Genetic correlation between ASD 3,199 brain imaging phenotypes from the Brain Imaging Genetics project and (B) genetic correlation between 14 brain imaging phenotypes nominally genetically correlated with ASD.
(DOCX)

**S5 Fig. Summary of the relationship between autism spectrum disorder (ASD) polygenic risk score (PRS) and anger recognition (PEITANG; Penn Emotion Identification Test correct responses to anger trials) across Philadelphia Neurodevelopmental Cohort (PNC) proband age groups.**
(DOCX)

**S1 Table. Phenotypes meeting suggestive threshold for significant prediction by polygenic risk for autism spectrum disorder in the young (YP), middle (MP), and adult (AP) proband groups.**
(DOCX)

**S2 Table. Spearman's correlation ($r^2$) and significance (p-value) between PEITANG and other measures recorded by the Penn Emotion Identification Test (PEIT).**
(DOCX)

**S3 Table. Traits significantly correlated ($r^2$ and correlation p) with PEITANG in the middle proband group of the Philadelphia Neurodevelopmental Cohort and at least nominally associated with polygenic risk (PRS) for autism spectrum disorder (PRS $R^2$, z-score, and PRS p).**
(DOCX)

**S4 Table. Samples size and SNP-based observed-scale heritability ($h^2$) for genome-wide association study summary statistics used in this study.**
(DOCX)

**S5 Table. Phenotypes associated with polygenic risk for autism spectrum disorder based on perceived behavior of middle age group (11 to 17 years old) informants from the**

**Philadelphia Neurodevelopmental Cohort.**
(DOCX)

## Acknowledgments

The authors thank the Philadelphia Neurodevelopmental Cohort (PNC) data collection and curation team.

## Author Contributions

**Conceptualization:** Frank R. Wendt, Renato Polimanti.

**Data curation:** Frank R. Wendt, Carolina Muniz Carvalho, Gita A. Pathak.

**Formal analysis:** Frank R. Wendt.

**Funding acquisition:** Renato Polimanti.

**Investigation:** Frank R. Wendt, Carolina Muniz Carvalho, Gita A. Pathak, Joel Gelernter, Renato Polimanti.

**Resources:** Joel Gelernter.

**Supervision:** Renato Polimanti.

**Writing – original draft:** Frank R. Wendt, Renato Polimanti.

**Writing – review & editing:** Frank R. Wendt, Carolina Muniz Carvalho, Gita A. Pathak, Joel Gelernter, Renato Polimanti.

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
