## [Decision Letter · Decision Letter 0]

14 Jul 2020

Dear Dr Polimanti,

Thank you very much for submitting your Research Article entitled 'Polygenic risk for autism spectrum disorder associates with anger recognition in a neurodevelopment-focused phenome-wide scan of unaffected youths from a population-based cohort' to PLOS Genetics. Your manuscript was fully evaluated at the editorial level and by independent peer reviewers. The reviewers appreciated the attention to an important problem, but raised some substantial concerns about the current manuscript. Based on the reviews, we will not be able to accept this version of the manuscript, but we would be willing to review again a much-revised version. We cannot, of course, promise publication at that time.

If you decide to revise the manuscript for further consideration at PLOS Genetics, please aim to resubmit within the next 60 days, unless it will take extra time to address the concerns of the reviewers, in which case we would appreciate an expected resubmission date by email to plosgenetics@plos.org.

[LINK]

We are sorry that we cannot be more positive about your manuscript at this stage. Please do not hesitate to contact us if you have any concerns or questions.

Yours sincerely,

Scott M. Williams

Section Editor: Natural Variation

PLOS Genetics

Scott Williams

Section Editor: Natural Variation

PLOS Genetics

Reviewer's Responses to Questions

**Comments to the Authors:**

Reviewer #1: Wendt et al conducted very interesting study which examined possible association between polygenic risk score of autism spectrum disorder and various neurodevelopmental domains using large sample size (N=4,309). The manuscript is concise and the results were statistically robust. I believe that the conclusion drawn is very attractive for readers.

Major comments:

Author summary

The last sentence sounds a bit strong since the subjects analyzed in this study are lack of ASD diagnosis.

Results:

1. The brief summary of results for measured neurodevelopmental domains should be described in the main text.

2. I do not deny in using Best-fit PRS methods, but it is known that this method may cause over-fitting. Although figure 3 clearly demonstrated that association of ASD PRS and PEITANG is robust and not a type I error, the p-value threshold and the number of SNPs analyzed in best-fit model should be described somewhere (main text or figure 1).

Discussion:

1. The authors hypothesized that it is difficult for ASD subjects to perceive positive emotion or neutral emotion. I believe that this hypothesis has been supported by many previous studies, so references should be cited.

2. The author also hypothesized that ASD children are familiar with anger because their caregivers have anger emotion for raising them. I do not agree with this since the target population has lack of ASD diagnosis. The authors would like to say that as they are probands for ASD subjects, they have more opportunities to see that their caregivers get angry with their siblings?

Methods:

The summary of measured neurodevelopmental domains and batteries should be described in the main text.

Minor points

1. In the abstract, please indicate if the p-values have been corrected for multiple testing or not in the abstract.

2. Ref 15 (Demontis D, Walters RK, Martin J, Mattheisen M, Als TD, Agerbo E, et al. Discovery Of The First Genome-Wide Significant Risk Loci For ADHD. bioRxiv. 2017) has already been published in Nature Genetics in 2018.

3. In supplemental tables, please add descriptions for abbreviations, such as GWS, correlation P, PRS P.

Reviewer #2: The authors performed a phenome-wide association study the of ASD PRS on 491 neurodevelopmental phenotypes in 4,309 probands in the Philadelphia Neurodevelopmental Cohort. The results were stratified by age group and it is noted that the these individuals lack an ASD diagnosis. An positive association with between the ASD PRS and PEITANG, a quantitative trait that measures variability in the recognition of angry emotion was observed. This association was follow up through the analysis of additional brain-imaging and psychiatric polygenic scores. The aim of these analyses is to demonstrate that evidence of an ASD-specific polygenic signal by adjusting for additional PRS. This is an interesting approach, although difficult to follow at times and would benefit from the flow chart in the supplement moved to the main text.

Overall, I found this to be an interesting follow up study from Grove et al. for understanding the contribution of the autism polygenic score to behavior and psychometric traits in the general population. Overall, the manuscript is well-written and will be of interest to autism researchers studying common variation. My main criticism is that the scope of the research is narrow. For example, it is unclear why the investigators limited the PheWAS to one neurodevelopmental/psychiatric polygenic score (ASD PRS) in the NPC cohort. In my opinion, a battery of polygenic scores in the main PheWAS analysis to understand polygenic associations with NPC measurements would have been worthwhile.

Questions/Comments

1. Ascertainment into the PNC cohort should be in mentioned in the main text prior to the dissemination of the results. While noting the lack of ASD diagnoses is helpful, it should be more clearly stated that these patients are not enriched for any clinical diagnoses and more representative of adolescents from the general population.

2. The total number of study-wide tests is not clearly stated in the manuscript. For each PRS, it seems it all the different p-value thresholds and all were included in the PheWAS. It’s not clear why this was selected over choosing a single PRS a priori as reported in Zheutlin et al.. Similar to SCZ, ASD is highly polygenic trait and unlikely to benefit in performance from p-value thesholding:

Zheutlin AB, Dennis J, Karlsson Linnér R, Moscati A, Restrepo N, Straub P, et al. Penetrance and Pleiotropy of Polygenic Risk Scores for Schizophrenia in 106,160 Patients Across Four Health Care Systems. Am J Psychiatry 2019;176:846–55.

Can the authors justify their motivation for including PRS generated from all p-value threshold in the analysis and clarify this in the methods?

3. The authors note that the ASD PRS association with PEITANG did not replicate in young probands. Can the heritability of PEITANG be tested the young proband group and compared to the older samples using snp-based approaches like GCTA?

4. Was male-sex associated with PEITAND and were any sex-PRS interactions observed?

5. How many ASD samples are present in the NPC cohort? If samples are present in the NPC, what is the PEITAND distribution in ASD cases compare to the remainder of the cohort.

6. In the discussion, the authors describe the association between the ASD and PEITANG in the context of the ASD phenotype. However, a stronger association with PEITANG was observed with the schizophrenia PRS, indicating that increased recognition of angry faces may be a common feature of the genetic risk for neuropsychiatric conditions and not ASD-specific. Can the authors comment on this alternative explanation in their manuscript? The discussion of systematizing reads as conjecture without evidence and should be reduced in the text.

7. Are there reports of PheWAS of the ASD PRS in the UK Biobank? If so, this would be an obvious study of an adult population for comparison with the results in adolescents here.

**Have all data underlying the figures and results presented in the manuscript been provided?**

Reviewer #1: Yes

Reviewer #2: Yes

PLOS authors have the option to publish the peer review history of their article (what does this mean?). If published, this will include your full peer review and any attached files.

Reviewer #1: **Yes: **Nagahide Takahashi

Reviewer #2: No

---

## [Decision Letter · Decision Letter 1]

8 Aug 2020

Dear Dr Polimanti,

We are pleased to inform you that your manuscript entitled "Polygenic risk for autism spectrum disorder associates with anger recognition in a neurodevelopment-focused phenome-wide scan of unaffected youths from a population-based cohort" has been editorially accepted for publication in PLOS Genetics. Congratulations!

Yours sincerely,

Scott M. Williams

Section Editor: Natural Variation

PLOS Genetics

Scott Williams

Section Editor: Natural Variation

PLOS Genetics

Comments from the reviewers (if applicable):

Reviewer's Responses to Questions

**Comments to the Authors:**

Reviewer #1: The authors have addressed all the comments the reviewer raised.

Reviewer #2: My concerns with this manuscript have been addressed by authors. Their responses to the reviewers questions were well reasoned and stated clearly. I find this version manuscript nicely written and an enjoyable read.

**Have all data underlying the figures and results presented in the manuscript been provided?**

Reviewer #1: Yes

Reviewer #2: Yes

PLOS authors have the option to publish the peer review history of their article (what does this mean?). If published, this will include your full peer review and any attached files.

Reviewer #1: **Yes: **Nagahide Takahashi

Reviewer #2: No

**Data Deposition**

http://datadryad.org/submit?journalID=pgenetics&manu=PGENETICS-D-20-00575R1

**Press Queries**

---

## [Editor Report · Acceptance letter]

8 Sep 2020

PGENETICS-D-20-00575R1 

Polygenic risk for autism spectrum disorder associates with anger recognition in a neurodevelopment-focused phenome-wide scan of unaffected youths from a population-based cohort 

Dear Dr Polimanti, 

We are pleased to inform you that your manuscript entitled "Polygenic risk for autism spectrum disorder associates with anger recognition in a neurodevelopment-focused phenome-wide scan of unaffected youths from a population-based cohort" has been formally accepted for publication in PLOS Genetics! Your manuscript is now with our production department and you will be notified of the publication date in due course.

With kind regards,

Matt Lyles

PLOS Genetics

On behalf of:
